# The Infectious Dose Shapes *Vibrio cholerae* Within-Host Dynamics

Aaron Nicholas Gillman,[a,b] Anel Mahmutovic,[a] Pia Abel zur Wiesch,[a,c,d,e] Sören Abel[a,b,c,e]

[a]Department of Pharmacy, Faculty of Health Sciences, The Arctic University of Norway, Tromsø, Norway
[b]Department of Veterinary and Biomedical Sciences, The Pennsylvania State University, University Park, Pennsylvania, USA
[c]Centre for Molecular Medicine Norway, Nordic EMBL Partnership, Oslo, Norway
[d]Department of Biology, The Pennsylvania State University, University Park, Pennsylvania, USA
[e]Huck Institutes of the Life Sciences, The Pennsylvania State University, University Park, Pennsylvania, USA

**ABSTRACT** During infection, the rates of pathogen replication, death, and migration affect disease progression, dissemination, transmission, and resistance evolution. Here, we follow the population dynamics of *Vibrio cholerae* in a mouse model by labeling individual bacteria with one of >500 unique, fitness-neutral genomic tags. Using the changes in tag frequencies and CFU numbers, we inform a mathematical model that describes the within-host spatiotemporal bacterial dynamics. This allows us to disentangle growth, death, forward, and retrograde migration rates continuously during infection. Our model has robust predictive power across various experimental setups. The population dynamics of *V. cholerae* shows substantial spatiotemporal heterogeneity in replication, death, and migration. Importantly, we find that the niche available to *V. cholerae* in the host increases with inoculum size, suggesting cooperative effects during infection. Therefore, it is not enough to consider just the likelihood of exposure (50% infectious dose) but rather the magnitude of exposure to predict outbreaks.

**IMPORTANCE** Determining the rates of bacterial migration, replication, and death during infection is important for understanding how infections progress. Separately measuring these rates is often difficult in systems where multiple processes happen simultaneously. Here, we use next-generation sequencing to measure *V. cholerae* migration, replication, death, and niche size along the mouse gastrointestinal tract. We show that the small intestine of the mouse is a heterogeneous environment, and the population dynamic characteristics change substantially between adjacent gut sections. Our approach also allows us to characterize the effect of inoculum size on these processes. We find that the niche size in mice increases with the infectious dose, hinting at cooperative effects in larger inocula. The dose-response relationship between inoculum size and final pathogen burden is important for the infected individual and is thought to influence the progression of *V. cholerae* epidemics.

**KEYWORDS** founder population size, moment closure, particle swarm optimization, stochastic spatiotemporal population dynamics, dose response, *Vibrio cholerae*, cholera, host model

During bacterial infections, pathogens invade the host and colonize various niches where the bacteria grow and die, which results in constant changes of the pathogen burden. Pathogens can also migrate within the host, moving to new regions of the body or changing the composition of subpopulations by migrating between colonized regions. The rates with which these processes take place depend on host-pathogen interactions. For example, immune effectors may increase the death rate of the pathogens without having a substantial impact on the replication rate, or, conversely, nutrient limitation can decrease the replication rate without perturbing the death rate

Address correspondence to Sören Abel, soren.abel@uit.no.

The authors declare no conflict of interest.

(1, 2). Information about replication and death rates can illuminate the specific nature of host-pathogen interactions (1, 3–5). To optimize treatment strategies, we need a better understanding of these processes. Just as there are vulnerable molecular targets that can be exploited for fighting pathogens, targeting treatment to small or subdivided populations can facilitate elimination of the pathogen from the host (4).

Although understanding these rates is a prerequisite for rational anti-infective treatment, it is difficult to measure replication, death, and migration rates experimentally, and they are often set to a range of biologically plausible values (3, 6, 7). Furthermore, these data can be difficult to obtain, as their measurement often requires sacrificing the host, thereby prohibiting continuous measurements within a single animal and limiting the number of data points due to ethical considerations.

In this work, we employ *Vibrio cholerae*, a noninvasive pathogen that infects the gastrointestinal (GI) tract (8). In addition to causing a substantial disease and mortality burden, the properties of *V. cholerae* infection make it an exceptional model organism for studying population dynamics, as it exists in confined well-characterized compartments with measurable and controllable inputs and outputs. Recent outbreaks have highlighted the need to further understand the factors critical for colonization and infection by *V. cholerae* (9, 10). For example, in human studies, disease severity and the probability of infection were associated with the magnitude of exposure (11). However, a better understanding of the effects of inocula on population dynamics is needed for modeling disease progression. Specifically, the relationship between inoculum size and transmission, and thus, pathogen burden of the infecting and infected host, is important for understanding outbreaks (12).

Current models for *V. cholerae* colonization are primarily formed from physiological observations and mathematical predictions based on research done with mutants and fluorescence studies (6, 13). Quantification of bacterial CFU numbers and microscopy are useful tools for understanding infection and virulence. Use of competitive indices has elucidated the phenomena of hyperinfectivity and the contribution of motility to colonization (14, 15). Fluorescent microscopy has characterized microcolony formation within crypts and shedding of microbiofilms to the lumen (16). However, these studies are limited to measuring a single or a few factors at a time by using mutants or fluorescence labeling. While these studies are useful in identifying key factors of pathogenesis, when applied to model development, substantial assumptions are made to account for factors such as bacterial growth and death rates in multiple compartments.

Recent developments in modeling other pathogens have used the WITS method (wild-type isogenic tag strains) to study bacterial population dynamics within the host (3). The WITS method introduces artificial genetic variation by inserting a marker sequence in the genome. Bacterial replication, death, and migration rates can be inferred by quantifying the variation in the genetic composition of the bacterial population (3). However, these studies typically employ only a few distinguishable markers, which limits the accuracy in the estimated rates. Advancements in next-generation sequencing have allowed development of new techniques such as STAMP (sequence tag-based analysis of microbial populations) (17). STAMP uses randomly generated tag or barcode sequences inserted into a fitness-neutral region of the genome to create a microbial population where each cell contains one of many unique tags. This allows distinguishing between subpopulations of bacteria, the proportions of which are used to calculate the founding population size ($N_B$) in a single experiment (17). The founding population size can be interpreted as the number of the bacteria that colonized the host, i.e., survive random death events and gave rise to the current bacterial population (Fig. 1).

In this study, we use our recently developed method, RESTAMP (rate estimate by sequence-tag analysis of microbial populations), to investigate the population dynamics of *V. cholerae* in an infant mouse model host (18). We employ a library of bacteria tagged with ~500 unique, 30-bp-long tags (see Materials and Methods). In addition to classic determination of the total population size (census population) at the time of

## A.

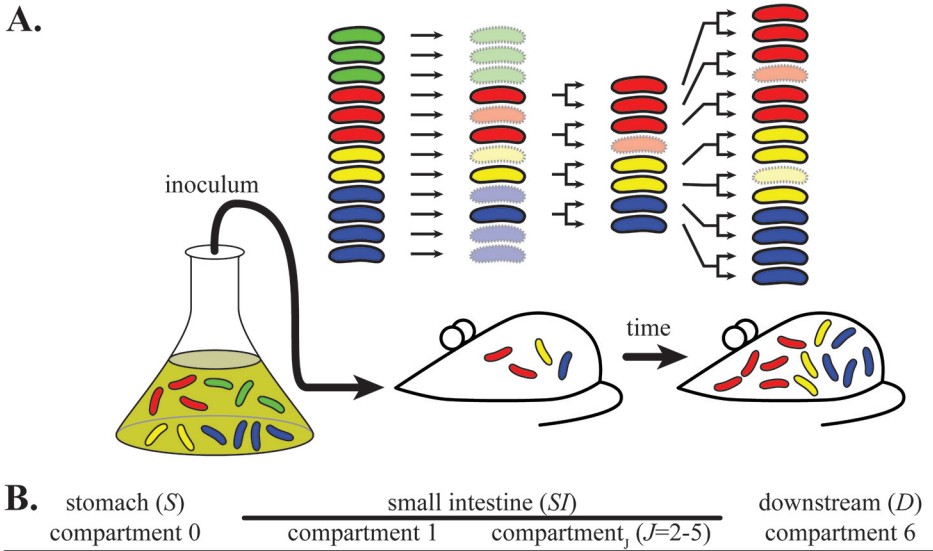

## B.

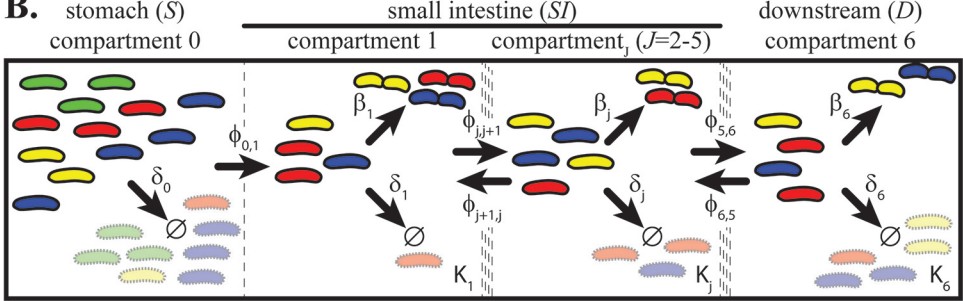

**FIG 1** Monitoring tagged bacterial populations in experiments and models. (A) Schematic representation of the experimental setup. Bacteria are labeled with heritable genetic tags (exemplified by red, green, yellow, and blue color). Mice are inoculated from a starting population grown *in vitro*. After infection, random death events (transparent bacteria) reduce the population size and tag diversity (e.g., disappearance of the green bacteria). Colonizing bacteria can then replicate, and if they grow without competition, tag frequencies do not change. If bacteria die, the tag frequencies change. The difference between replication rate and death rate determines the net growth rate, i.e., the change in population size over time. (B) Illustration of the population dynamical model. The GI tract is split into six compartments (separated by dotted lines), starting with the stomach ($S$, compartment 0), five compartments in the small intestine ($SI$, compartments 1 to 5), and a final composite downstream compartment accounting for all downstream sections, like the cecum, large intestine, and secreted bacteria ($D$, compartment 6). Arrows indicate processes that affect the population dynamics. Each compartment has a carrying capacity ($K$), death rate ($\delta$), replication rate ($\beta$), and forward and retrograde migration rate ($\varphi$), except for the stomach, where we only consider death and forward migration rate. Bacteria with a central constriction indicate replication; bacterial death is indicated by Ø.

sampling, these tags allow us to infer the founding population size. We combine these experimental data with a mathematical model that couples stochastic dynamics with population genetics, which enables us to determine the replication, death, and migration rates of *V. cholerae* by combining data from several hosts. To accomplish this, we connect hitherto separate studies on stochastic population dynamics (3–5, 19) and population genetic models on within-host infection dynamics (17, 20–22).

We monitor the change in replication, death, and migration rates within five sections of the small intestine over 24 h and find a surprising spatiotemporal heterogeneity. Notably, we find large retrograde migration rates that drive recolonization of *V. cholerae* as infection progresses. In addition, our model pinpoints distal regions of the small intestine to be more conducive for bacterial proliferation. Furthermore, we investigate the impact of the inoculum size on the severity of the disease and the founding population ($N_B$). We find a largely linear correlation between the infectious dose and $N_B$ (4.9% of the inoculum survives) above the 50% infectious dose (ID$_{50}$; ~10 to 100 *V. cholerae* cells). Surprisingly, our model finds that the bacterial load (CFU) cannot be explained if all parameters of the model are the same for all inoculum sizes. Instead, the carrying capacity of the host compartments needs to increase with increasing inoculum size for a good model fit. We therefore conclude that the infectious dose changes

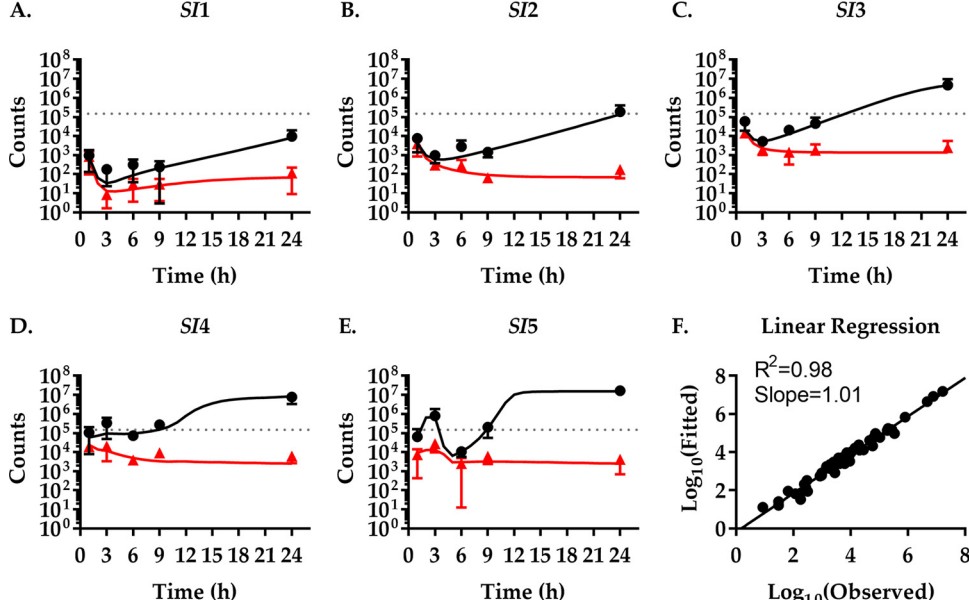

**FIG 2** Experimentally observed and modeled population dynamics of *Vibrio cholerae*. (A to E) Mice were infected with an inoculum of 1.48 × 10⁵ CFU *V. cholerae* per mouse (dotted line) and sacrificed at the indicated times to isolate bacteria and determine bacterial CFU numbers (black dots) and $N_B$ (red triangles) from five different sections of the small intestine (*SI*1 to -5). Data show the mean from three mice ±SD. A mathematical model of the population dynamics was fitted to the experimental values. Black (CFU) and red ($N_B$) lines represent continuously fitted model values. (F) Correlation between experimentally determined (observed) and modeled (fitted) values for CFU numbers and $N_B$.

the population biology of *V. cholerae* infections and suggest that the pathogen cooperates to increase its niche in the host.

## RESULTS

**Fitting a multicompartmental model to experimental data.** To investigate the population dynamics of *V. cholerae* infections, infant mice were orogastrically infected by gavage with a dose of 1.48 × 10⁵ CFU with the described experimental setup (Fig. 1A). The inoculum consisted of a library of tagged *V. cholerae* where each bacterium contains one of ~500 unique, 30-bp-long tag sequences that were inserted in a fitness-neutral position in the genome (17). Postinfection, we sacrificed the animals at different time points (1 h, 3 h, 6 h, 9 h, and 24 h) and isolated *V. cholerae* from different sites of the gastrointestinal tract. Starting from the stomach, we divide the small intestine, which is believed to be the major location of *V. cholerae* colonization and pathogenicity, into five subsequent and equal-length sections (*SI*1 to -5) (13, 23). From the isolated bacteria, the actual bacterial load (or total population size) at the time of harvest was determined by counting CFU. In addition, the founding population size ($N_B$) was determined by next-generation sequencing of the tag sequences and comparing the tag frequencies in the sample with the inoculum using STAMP (17).

Already at 1 h postinfection, we find a high pathogen burden in the entire small intestine (Fig. 2). This is followed by a steep decline in CFU numbers between 1 h and 3 h in the most proximal three sections (*SI*1 to -3), reducing the pathogen population by an average of 87%. Starting at 6 h, the bacterial burden steadily increases. The same pattern of increase, decrease, and increase is observed in the distal sections (*SI*4 and -5), albeit with a several-hour delay and more pronounced in *SI*5 than in *SI*4.

The $N_B$ values initially follow the CFU changes at a lower count. This indicates that the initial colonization is dominated by a rapid spread and death of the inoculum with limited replication of the pathogen. However, as CFU numbers increase the $N_B$ values remain relatively constant. This indicates that the number of founder bacteria remains constant, i.e., no new-tagged bacteria migrate here. However, the founders that are

mSystems®

present replicated to give rise to offspring with the same tags. An exception is the $SI1$ section where a slight increase in $N_B$ from 6 h to 24 h is observed, which is consistent with pathogen migration to this section, increasing tag diversity and thereby $N_B$.

In general, both CFU and $N_B$ values are relatively low in the proximal small intestine and increase toward the distal sections, which appear more permissive to *V. cholerae* colonization and growth (as reflected by high $N_B$ and CFU values, respectively). The majority of *V. cholerae* (83%) resides in the $SI4$ and -5 compartments at 24 h. The population dynamics early during infection appears to be dominated by migration events. Peristaltic flow and bile, a chemorepellent to *V. cholerae*, may drive migration through the gut and toward the distal regions (24). In contrast, bacterial replication and death events appear to play a greater role later during the infection.

Already from this qualitative analysis, it is apparent that the small intestine is a heterogeneous environment. Changes in *V. cholerae* burden and founding populations over time are different in the sections of the GI tract, indicating that the bacteria are not freely floating within the gut and that populations behave differently in each part of the intestine. However, without a quantitative understanding it is difficult to disentangle the different processes that dictate these behaviors.

To gain such a quantitative understanding, we built a model to capture the dynamics during colonization and spread through the host (Fig. 1B) and applied it to the data set to continuously calculate CFU and $N_B$ values (Fig. 2, black and red lines). We consider three different types of rates in each compartment: bacterial replication, bacterial death, and migration between compartments. This model was then fitted to our experimental data of the total population size (CFU numbers) and founding population size ($N_B$) to infer the magnitude of these rates. As an intuitive explanation, bacterial death reduces the tag diversity in the population. In contrast, bacterial replication, in the absence of competition, leaves the tag frequencies unchanged. For example, if bacteria neither replicate nor die, we would observe a population that has both a constant total size and constant tag frequencies. If a constant total population size is the result of equal replication and death rates, the tag diversity would decrease, and the magnitude of this decrease would inform us of the rates. Additional changes in bacterial tag frequency that leave tag diversity unchanged but change individual tag frequencies, for example, toward uniformity, allow us to distinguish the magnitude and source of bacterial migration. Thus, the total population size, together with tag frequencies (measured as $N_B$), enable us to estimate the replication, death, and migration rates in tagged populations.

We have triplicates of 50 observations, ideally allowing for fitting models with up to 50 parameters. However, a formal analysis of parameter identifiability is computationally difficult; therefore, we aimed to minimize model complexity and, thus, the number of parameters. A minimal mechanistic model of the within-host spatiotemporal dynamics of *V. cholerae* with 30 parameters and constant replication, death, forward, and retrograde migration rates is not sufficient to capture the variation in the experimental data. By stepwise increases in model complexity to 40 parameters, we find that a model with time-dependent migration rates is necessary to fit the experimental data well and make accurate predictions (see Fig. S1 in the supplemental material). Given this model, the spatiotemporal variation in the CFU and $N_B$ data is captured to a high degree of accuracy, as measured by the coefficient of determination ($R^2 = 0.98$) and the slope of the linear regression (slope = 1.01) for the fitted model output and the experimental data (Fig. 2F).

**Rates of individual processes.** The model we fitted to the experimentally determined CFU and $N_B$ data can be used to investigate the population dynamics of *V. cholerae* in the small intestine of mice in detail. With our model, we can monitor replication, death, carrying capacity, forward migration, and retrograde migration events. To get an understanding of the dynamic activity of *V. cholerae*, we determined the rates from our fitted model. The rates denote the time per event for individual bacteria and not for the population. For example, we see substantial migration rates from $SI1$ to $SI2$

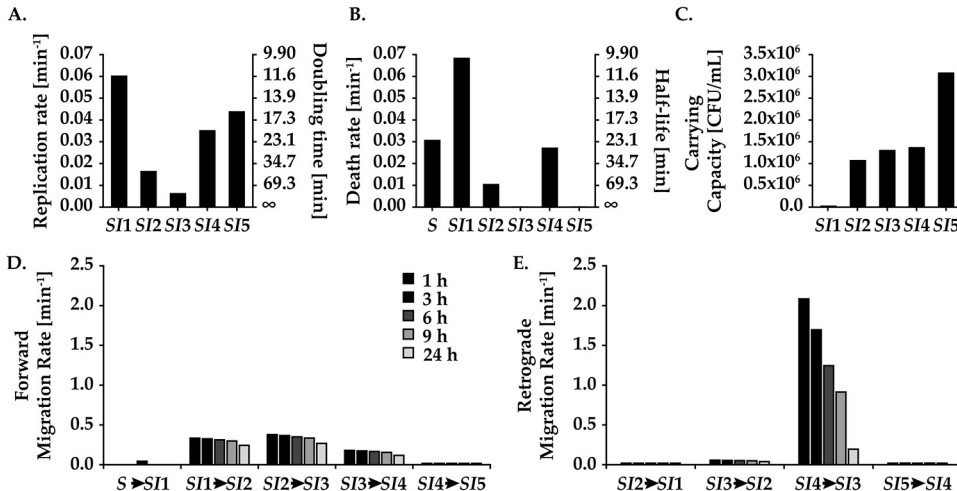

FIG 3 Small intestine population dynamics of *Vibrio cholerae* reduced into division, death, carrying capacity, and migration rates. The values for the rates (*y* axis) were acquired from our fitted model (Fig. 2) for replication (A) and death (B). On the right, the *y* axis denotes doubling time and half-life of bacterium, respectively. (C) The carrying capacities of the SI segments. (D and E) Forward (D) and retrograde (E) migration events over time from 1 h (black bar) to 24 h (light gray bar).

and *SI*2 to *SI*3, which drives the bacterial population to the distal end of the small intestine in a very short time (Fig. 3). Although the *SI*4 to *SI*5 migration rates are lower, the CFU count is large enough (Fig. 3D and E) to make migration events from *SI*4 to *SI*5 significantly more likely than either replication or death events. This is due to the forward migration rates incrementally increasing the total bacterial population in the subsequent compartments. Hence, already at 1 h postinfection the population has been driven to the distal ends of the intestine by a large forward force. Until 3 h after inoculation, the dynamics are primarily comprised of migration events to and from *SI*4 and *SI*5 and retrograde migration events from *SI*4 (Fig. 3D and E). Overall, retrograde migration effects become negligible after 9 h, with forward migration decreasing slowly to 24 h. The large retrograde migration rates were unexpected from a biological point of view, as we would expect the strong peristaltic forces of the intestine to move the population forward.

After a phase dominated by migration, a rapid turnover of cells ensues predominantly in *SI*1 and *SI*4, with the magnitude of turnover greatest in *SI*4. Net bacterial growth remains low in *SI*1 due to a low carrying capacity, high death rates, and an overall substantial forward migration rate. We see a progressively decreasing replication rate in the proximal regions (*SI*1 to -3) (Fig. 3A). Taken together with the substantial forward migration rates, this resulted in a decreased bacterial burden in the proximal regions (Fig. 3A to C). Consequently, this makes bacterial growth less likely to occur in the proximal regions (*SI*1 to -3). In contrast, the bacterial load is significantly larger in the distal regions (*SI*4 and -5), where we observe an overall net positive growth rate (Fig. 2A). Bacteria residing in *SI*5 grow with the highest net rate due to a high individual replication rate, low death rate, and large total bacteria population (Fig. 3A to C). Death rates in *SI*3 and *SI*5 were minimal, <0.001 min⁻¹, with nearly all loss of bacteria in these compartments from migration. In summary, migration events dominate shortly after infection, followed by heterogeneous replication and death dynamics dominating in the distal regions before significant growth begins in proximal regions.

The rates determined by our fitted model were derived from experiments with a single inoculum size. Therefore, the determined rates may only be valid for the specific experimental setup and inoculum size. To exclude this, we next tested the predictive power of our model.

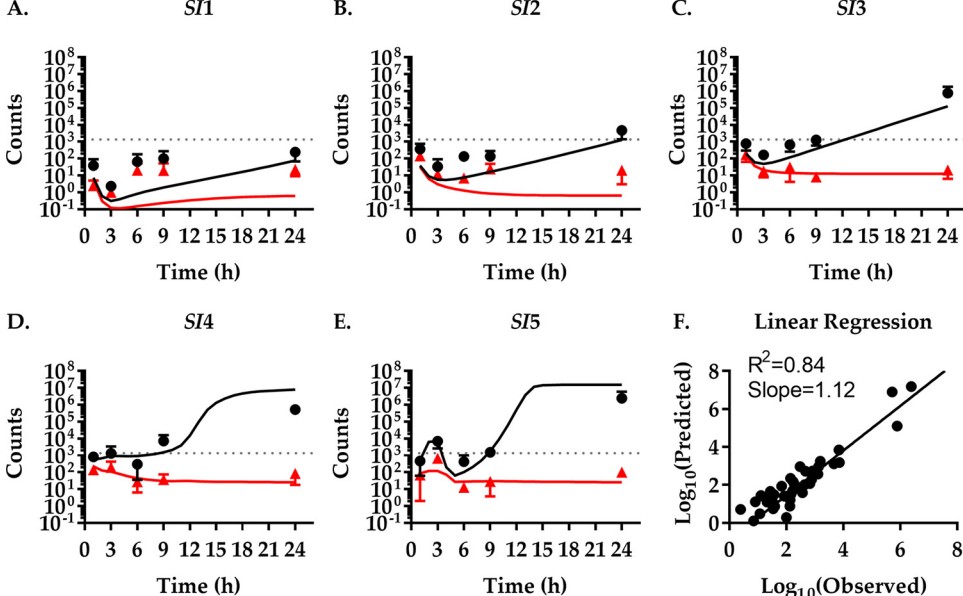

**FIG 4** High predictive power of the population dynamics at a lower inoculum. Comparison of the model fitted to experimental data from Fig. 2 with experimental results for infections with a low inoculum. (A to E) Simulation of infections with a low inoculum of $1.37 \times 10^3$ CFU *V. cholerae* per mouse (dotted line). CFU (black lines) and $N_B$ (red lines) over time in five different compartments of the small intestine (*SI*1 to -5) were predicted by a mathematical model fitted to experimental data of mice infected with an inoculum of $1.48 \times 10^5$ CFU *V. cholerae* per mouse (Fig. 2). The quality of predictions was tested experimentally by infecting mice with the low inoculum of $1.37 \times 10^3$ CFU *V. cholerae* per mouse and sacrificing them at indicated times to isolate bacteria and determine bacterial CFU numbers (black dots) and $N_B$ (red triangles). Data show the mean of three mice ±SD. (F) Correlation between experimentally determined (observed) and modeled (predicted) values for CFU and $N_B$.

**The predictive power of our population dynamics model.** To test the predictive power of our fitted model of *V. cholerae* infection, we used the rates determined before to predict the population dynamics at a lower inoculum. To confirm the quality of the predictions, we performed an additional experiment with an inoculum 2 orders of magnitude lower, $1.37 \times 10^3$ CFU/mouse. These data (Fig. 4) show a pattern similar to that of infections with the higher inoculum (Fig. 2). A rapid distribution of *V. cholerae* occurs within the first hour postinfection. This is followed by a decrease in total CFU number (by 81%) in the small intestine in the proximal regions *SI*1 to -3 before the pathogen population increases steadily. The *SI*4 and -5 regions follow this pattern with a several-hour delay. Again, the distal small intestine is more permissive to *V. cholerae* colonization and growth, with the majority (79%) of bacteria being located in the *SI*4 and -5 compartments at 24 h. However, the total pathogen numbers only reach about 1/10 of the total pathogen burden in mice infected with the higher infectious dose.

The $N_B$ initially mirrors the change in CFU only to remain largely constant once *V. cholerae* numbers begin to increase after 6 h. Interestingly, the increase in bacterial numbers in *SI*5 and, to a lesser extent, *SI*4 up until 3 h is accompanied by an increase of $N_B$, indicating that it is caused, at least partially, by bacterial migration. Similarly, $N_B$ increases in *SI*1 between 3 h and 6 h after being nearly depleted of *V. cholerae* ($N_{B, 3h} \leq$ 1), again indicating bacterial migration. The change in the CFU and $N_B$ data due to the lower inoculum is well captured by our model, measured by the coefficient of determination ($R^2 = 0.84$) and the slope of the linear regression (slope = 1.12) of the correlation between the fitted model output and experimental data (Fig. 4F).

**Predicting the population dynamics for the entire small intestine.** Our model is comprised of sections of the GI tract, for each of which CFU and $N_B$ data are modeled separately. We performed an additional experiment with an inoculum of $8.83 \times 10^5$ CFU/mouse. However, instead of measuring sections of the small intestine separately, CFU and $N_B$ values of the entire small intestine were determined (Fig. 5). At the same time, we convert the output of our fitted compartmental model to predict the

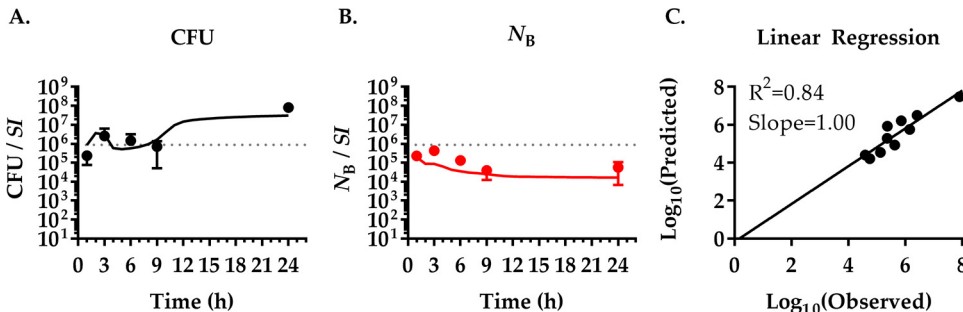

**FIG 5** Colonization of *V. cholerae* across the small intestine. Mice were infected with a high inoculum of $8.83 \times 10^5$ CFU *V. cholerae* per mouse (dotted line) and sacrificed at indicated times to isolate bacteria and determine bacterial CFU (black dots) (A) and $N_B$ (red dots) (B) from the entire small intestine. Data show the means from three mice ±SD. CFU (black lines) (A) and $N_B$ (red lines) (B) values continuously predicted by the mathematical model fitted to experimental data from Fig. 2. (C) Correlation between experimentally determined (observed) and modeled (predicted) values for CFU and $N_B$.

population dynamics of the whole small intestine. The CFU data of the whole small intestine mirror what we observed in Fig. 2 and 4 for the *SI4* and *SI5* compartments. This was expected, as these compartments comprise a majority of the bacterial burden and therefore dominate the overall bacterial dynamics. This shows that our model can also accurately predict the population dynamics in the small intestine as a whole.

**The pathogen burden depends on the inoculum size.** While our model predictions for the compartmental model and the whole intestine model are generally good, at 24 h the predicted CFU estimates deviate from the experimental values by almost 1 order of magnitude (Fig. 4 and 5). With inocula that are smaller than the inocula used in the training data set, we overestimate the bacterial burden. With inocula that are larger than the inocula used in the training data set, we underestimate them. Our model assumes that replication rates, death rates, and carrying capacities are independent of the size of the inoculum, i.e., *V. cholerae* cells replicate and die independently of each other. Thus, our model does not include competition and cooperativity in the pathogen population and host-pathogen interactions. However, the CFU number discrepancies at 24 h suggest that these biological mechanisms are important to fully capture the infection process at late time points. To test if the inoculum size influences the infection, we generated a dose-response curve for both CFU and $N_B$ values in the small intestine 24 h postinfection. We infected 93 mice with doses of *V. cholerae* ranging from very low (30 CFU) to very high ($7.8 \times 10^7$ CFU). The experimental data show that the dose that infects 50% of the mice ($ID_{50}$) was 40 CFU ($P < 0.05$, Poisson model), consistent with previously published results (25). At inocula of 30 CFU, when infected (three out of nine animals), we observed a mean bacterial burden of $3.2 \times 10^5$ ($\pm 2 \times 10^5$ standard deviations [SD]). Up to inocula of ~$10^6$, the bacterial burden steeply increases with increasing doses. At higher inocula (>$10^6$ CFU) the bacterial burden reaches ~$10^8$ CFU/mouse ($1.4 \times 10^8$ to $8.2 \times 10^8$ CFU/mouse; 95% confidence interval [CI]), suggesting that the bacterial load at 24 h cannot exceed this number independent of the inoculum size.

Generally, the founding population ($N_B$) mirrors the bacterial burden (CFU). A linear relationship between founding population and inocula up to ~$10^6$ CFU is observed. The mean $N_B$ is 4.9% (95% CI, 2.4 to 6.9%; between infectious doses of $1.5 \times 10^2$ and $4.7 \times 10^6$ CFU) of the inoculum, indicating that a constant fraction of the inoculum can overcome bottlenecks and host defense mechanisms to successfully colonize the host. While the $N_B$ dose-response curve appears to saturate for higher inocula (Fig. 6), the data are insufficient to discriminate between a linear model and a nonlinear saturating model (Fig. S2).

Our model predicts a linear relationship between $N_B$ and inoculum (Fig. 6). It accurately captures the $N_B$ dose-response for inocula up to ~$10^6$ CFU/mouse. The predicted linearity is a consequence of neglecting the carrying capacity in modeling $N_B$ values [see "The founder population size ($N_B$)," below], which is a biological statement of

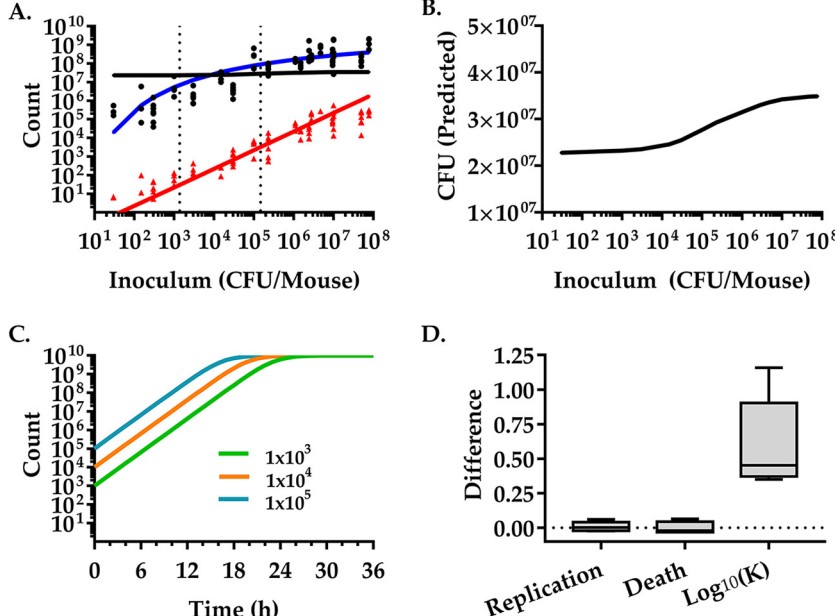

**FIG 6** Inoculum size strongly influences pathogen burden. Mice were infected with different inocula of *V. cholerae* and sacrificed at 24 h to isolate bacteria. (A) The *V. cholerae* CFU number (black dots) and $N_B$ (red triangles) were determined in the small intestine. Each data point represents one mouse (93 total). CFU (black line) and $N_B$ (red line) were predicted by a mathematical model fitted to experimental data of mice infected with an intermediate inoculum of $1.48 \times 10^5$ CFU *V. cholerae* per mouse (see Fig. 2). The blue curve shows the bacterial burden calculated by a Hill function of the inoculum size, $N(t) = K \times N(0)e^{rt}/[K+N(0)(e^{rt} - 1)]$ and $K = a \times N(0)/[b+N(0)]$ and was fitted to the logarithm of the experimental data using the Levenberg-Marquardt algorithm. The resulting values for the coefficients and the 95% confidence intervals are $a = 9.244$ (8.312, 10.18) and $b = 1.227$ (0.7051, 1.75) (adjusted $R^2 = 0.8613$). Dotted vertical lines represent model-fitted doses of $1.37 \times 10^3$ and $1.48 \times 10^5$ CFU. (B) Zoom-in of the predicted CFU numbers. The same data as the black line in panel A but on an enlarged, linear $y$ axis to show the nonlinear relationship between the inoculum size and the bacterial load at 24 h more clearly. (C) Schematic representation of logistic growth and the dependence of the bacterial load modeled at a given time on the inoculum ($10^5$, cyan; $10^4$, orange; $10^3$, green). (D) Differences between individual parameters (replication rates, death rates, and the $\log_{10}$ of the carrying capacity) derived from our model fit of the high-inoculum experiments (Fig. 2) to the parameters derived from the low inoculum experiment fits (Fig. S3). The dotted line is set to zero for reference.

ignoring competition, e.g., limited niche or resource availability. Other neglected effects, which could induce a saturation in the $N_B$ dose-response curve at high inocula, are time-dependent growth rates and death rates. This could be due to a change in the pathogen's capacity to overcome host defenses or microbiota over time.

The impact of neglecting these higher-order effects is most evident in the CFU dose-response curve, for which the model predictions poorly reflect the actual data (Fig. 6A, black line). In a model of logistic growth of bacteria in the GI tract, bacteria will reach a specific density at a given time, depending on the size of the inoculum (Fig. 6C). We fitted our model to the lower inoculum data (Fig. 4) to determine which parameters consistently change globally that could explain higher carrying capacities (Fig. S3). While the replication and death rates vary insignificantly, the carrying capacity consistently increases across all segments. Comparing the two model fits predicts an ~1 order of magnitude increase in carrying capacity by increasing the inoculum from $1.37 \times 10^3$ to $1.48 \times 10^5$ (Fig. 6D), which is consistent with the observed increase of the bacterial burden (Fig. 6A). We therefore refitted the model assuming that the carrying capacity depends on the inoculum size using a single-compartment logistic growth model for the bacterial burden, where $N(t) = K \times N(0)e^{rt}/[K+N(0)(e^{rt} - 1)]$ (Fig. 6A, blue solid line). The carrying capacity, $K$, was set to a Hill function of the inoculum size as $K = a \times N(0)/[b+N(0)]$. The division rate and death rate were set to the average division rate and death rate over the compartments of the small intestine, where $\beta = 0.0326$ min$^{-1}$ and

$\delta$ = 0.0213 min$^{-1}$. According to our model, the carrying capacity increases as the population density gets larger, with a theoretical maximum bacterial burden of $1.67 \times 10^9$. This is consistent with the bacteria modifying the environment to increase the available niche and make it more conducive for colonization and growth.

**From rates to population behavior.** While our modeling approach uncovers individual location-dependent rates, an intuitive understanding of the spatiotemporal population dynamics can be difficult. This is not only because of the sheer number of parameters involved. The migration rates are time dependent, and logistic bacterial growth depends not only on the replication rate but also on how close the population is to the carrying capacity. Thus, the population dynamics depends both on the magnitude of the rates and the bacterial burden. One example would be the persistently high migration rates from *SI*1 and *SI*2 (Fig. 3), which exert a limited influence on the dynamics of cells at later time points, because the majority of the bacterial burden lies elsewhere (Fig. 2). In other words, if there are no cells that can migrate, then the magnitude of the rate of migration does not matter. To quantify whether a dynamic process exerts influence on the overall population dynamics, we first scale each of the rates with the average bacterial burden for each compartment $j$ at time $t$, e.g., for the forward migration from $j$ to $j + 1$, $\Phi_{j,j+1}(t)<N_j(t)>$. Here, $\Phi_{j,j+1}(t)$ includes the exponential time-dependent term, as defined in Table 1, and is elaborated upon in "Modeling details," below. The product of a rate and a population size is typically referred to as a propensity in the context of stochastic dynamics (26). For example, the forward migration propensity from $j$ to $j + 1$ is 0 if either the forward migration rate or the bacterial burden is 0. Accordingly, the propensities are defined for replication, death, and retrograde migration events in compartment $j$ as $\beta_j<N_j(t)>(1 - <N_j(t)>/K_j)$, $\delta_j<N_j(t)>(1 - <N_j(t)>/K_j)$, and $\Phi_{j,j-1}(t)<N_j(t)>$, respectively (Table 1). The sum of event probabilities for the dynamical processes of each compartment comprising the small intestine at each time point is equal to one. Each of the propensities correspond to the individual terms in equation 3 for the average bacterial burden at time $t$ where the magnitude of a propensity determines how much it contributes to the rate of change of the bacterial burden. The propensity can therefore be interpreted as the tendency of a dynamic process to cause a change in the bacterial burden. Using this, we quantify whether a dynamic process exerts influence on the overall population dynamics while accounting for logistic growth, time dependence of the migration rates, and bacterial burden. To simplify comparisons across compartments where both the rates and the bacterial burdens vary over a wide range, we consider the relative propensities, e.g., the forward propensity divided by the sum of all propensities for forward migration. This is equivalent to the probability of executing a dynamic process (forward migration event, retrograde migration event, replication event, or death event) in the context of stochastic dynamics where both the time until the next event and the type of event are randomly chosen (26). A further simplification, which can aid in the interpretation of the probability of an event, is to consider it as a frequency. For example, a probability of a migration event of 0.99 with the probability of a division event of 0.01 would be approximately 99 migration events for every division event. This is an approximation, because the probabilities change slightly for each single event that executes. The probabilities for the different types of dynamic processes in our model are plotted versus time (Fig. 7A to D).

Several dynamic processes dominate, mostly concentrated in the distal parts of the small intestine (*SI*4 and *SI*5). This was expected, since the majority of the bacterial burden is in the distal part of the small intestine. Colonization can be grouped into three distinct phases, early (1 to 3 h), middle (3 to 9 h), and late (9 to 24 h). In the early phase, migration events dominate toward the distal regions, with migration into and out of the *SI*5 region. Retrograde migration to *SI*5 was unexpected, as the ileocecal valve is present at the junction. Growth during this phase is minimal in magnitude, concentrated to the *SI*5 region. The middle phase is characterized by high bacterial turnover in the *SI*4 region and a continued increase in growth in the *SI*5 region. Forward migration to the *SI*5 region begins to increase, while migration out of *SI*5 slows. During the late phase, bacterial growth and

**TABLE 1** Description of model parameters

| Variable[a] | Meaning | Comment(s) |
|---|---|---|
| $k$ | No. of uniquely identifiable tags | For the exptl data in this work, $k = 500$ |
| $i$ | Index denoting a specific tag | $i = \{1,2,\ldots,k\}$ |
| $j$ | Index denoting a specific compartment | Index $j$ runs from $\{0,1,\ldots,6\}$ where 0 labels the stomach compartment, 1–5 labels the S/1–5 compartments, and 6 labels the composite downstream compartment |
| $\Omega$ | Denotes the entire small intestine | Used instead of $j$, when the entire small intestine is modeled |
| $\langle N_j(t) \rangle$ | Mean population size in compartment $j$ at time $t$, referred to as the total population size in the text | For example, in this notation $N_0(0)$ corresponds to the population in the stomach at time of 0, which equals the inoculum size |
| $\langle n_{i,j}(t) \rangle$ | Mean population size with barcode $i$ in compartment $j$ | Referred to as the subpopulation size in the text |
| $f_{i,j}(t)$ | Proportion of cells with barcode $i$ in compartment $j$ at time $t$ | |
| $\langle N_{B_j}(t) \rangle$ | Mean founder population size in compartment $j$ at time $t$ | $N_B$ is replaced by NB for notational clarity when there are multiple subscript indices |
| $\langle N_B^{-1}{}_{i,j}(t) \rangle$ | Mean of the inverse of the founder population size for subpopulation $i$ in compartment $j$ at time $t$ | Auxiliary variable introduced for the sake of notational simplicity and ease of calculations |
| $\beta_j$ | Replication rate in compartment $j$ | Replication rate is defined as the inverse avg time it takes for a cell to divide |
| $\delta_j$ | Death rate in compartment $j$ | Death rate is defined as the inverse avg time it takes for a cell to die |
| $K_j$ | Carrying capacity in compartment $j$ | Carrying capacity in the compartments $j = \{1,2,\ldots,6\}$ is modeled as logistic growth in equation 3 |
| $\phi_{j,j\pm1}(t) = \phi_{j,j\pm1}(0)e^{-\gamma_{j,j\pm1}(t)}$ | Flow rate from compartment $j$ to compartment $j \pm 1$ with a relaxation constant $\gamma_{j,j\pm1} > 0$ | Time-dependent migration rates are necessary to get a good fit for the model to the exptl data (Fig. S1) |
| $\gamma_{j,j\pm1}$ | The relaxation coefficient | Determines how rapidly the migration rates decline with time. |
| $\varepsilon$ | Divergence measure between the model output and the exptl CFU and $N_B$ data | Given by equation 4 |

[a]Angle brackets denote averages over repetitions.

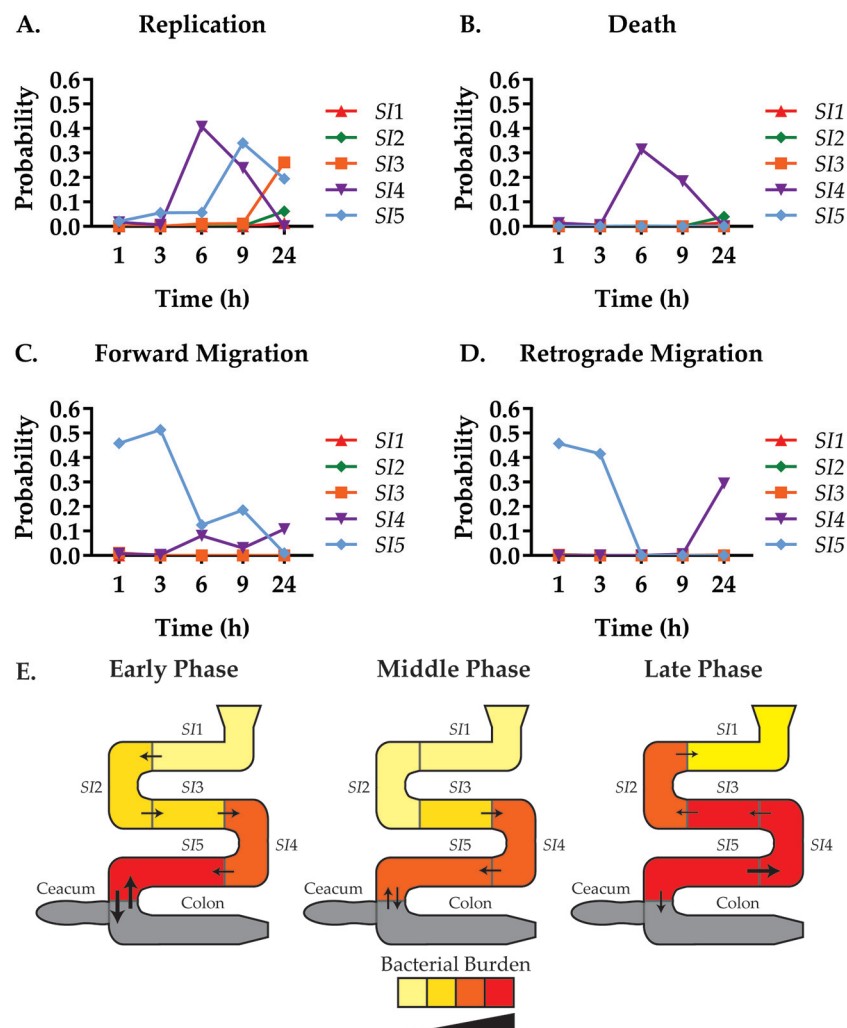

FIG 7 Illustrating the *Vibrio cholerae* dynamics in the small intestine of the mouse. (A to D) Event probabilities (y axis) were calculated as fractional propensities and are illustrated as a function of time (x axis) for the compartments *SI*1 (red), *SI*2 (green), *SI*3 (orange), *SI*4 (purple), and *SI*5 (cyan). The sum of event probabilities for the dynamic processes of each compartment comprising the small intestine at each time point is equal to 1. (E) Visual representation of the phases, early (1 to 3 h), middle (3 to 9 h), and late (9 to 24 h). Arrows and size represent approximate ratios of relative migration direction and magnitude, respectively. Color represents relative bacterial burden and growth for the CFU data in Fig. 2. The gray color indicates that experimental data are absent and that the cecum and colon were treated as a composite large intestine compartment in the model.

turnover begins in the proximal regions (*SI*1 to -3) in addition to retrograde migration from the distal regions (*SI*4). Thus, we observe both a time- and location-dependent heterogeneity of *V. cholerae* populations within the gut of the mouse.

## DISCUSSION

Understanding *in vivo* microbial behavior can aid in vaccine design and optimization of antimicrobial treatment regimens (3, 27, 28). In this study, we seek to understand the complex *in vivo* spatiotemporal population dynamics of *V. cholerae*. The rates with which *V. cholerae* migrates, replicates, and dies in distinct sections of the gut have not yet been determined. Determining those rates is of fundamental importance to understanding the specific nature of host-pathogen interactions (1, 3–5). We find a surprising heterogeneity and variation over time of *V. cholerae* dynamics. We also find that the increase of the infectious dose leads to a linear increase in initial colonization but a larger than expected increase in the bacterial burden over time.

mSystems®

Wild-type isogenic tagging (WITS), i.e., introducing traceable genetic variation in an otherwise identical population, is a powerful tool to investigate the spread of pathogens through their hosts (4, 19). Broadly speaking, these approaches can be classified (3) as population genetic (monitoring the composition of a pathogen population [21, 22]) or population dynamic approaches (monitoring the size of a pathogen population [4, 19]). We recently developed a joint approach, RESTAMP, by integrating the population genetic approach STAMP (17) with a stochastic population dynamic model and extended this approach by coupling it to the moment-closure framework (18). Here, we build on this work and employ a mathematical model that allows us to determine unique replication, death, and migration rates of *V. cholerae* in five compartments of the small intestine of the infant mouse.

A concern in model development is the risk of overfitting. We find that to fully capture the spatiotemporal heterogeneous population dynamics in the small intestine of the mouse, the model requires seven compartments and a total of 40 parameters using 50 observations. To mitigate the risk of overfitting, we employed two strategies. First, we stepwise increased the model complexity until a satisfactory model fit is achieved (see Fig. S1 in the supplemental material). Second, we conducted a series of experiments where we changed the inoculum size with sectional analysis (Fig. 4) and applied our prediction to the whole gut (Fig. 5). When lowering the inoculum or by analysis of the whole gut, the model still fits the data well (Fig. 4 and 5). This suggests that our parameter set is valid, but we cannot prove that the solution is unique. The model is 40-dimensional and computationally intense, but by using biological constraints on parameters, we can both ensure that our model is biologically plausible and reduce optimization problems. An advantage of our methodology is that there are no theoretical constraints on the maximum number of sequence tags. Increasing the number of tags decreases the uncertainty in $N_B$ measurements, buffers against loss of accuracy during death processes, and minimizes the number of animals used (17, 18). This makes it a powerful tool when considering validity and the 3R principles for ethical use of animals in testing: replacement, reduction, and refinement (29). We ultimately find a good fit ($R^2 = 0.98$) for a model that includes time-dependent migration rates between the small intestine compartments with unique death and replication rates for each compartment.

We observe heterogeneity of *V. cholerae* burden in the small intestine that is due to pronounced spatiotemporal differences in all of the population dynamical processes considered here, migration, replication, and death. Broadly speaking, an early phase of forward migration is followed by a proximal to distal wave of killing, while the final phase is characterized by retrograde migration and strong replication in the distal small intestine. Previous studies that investigated the changes in CFU numbers with either time or with infectious doses have observed similar bacterial burdens and the same trend of proximal-to-distal wave-like increase and decrease of the *V. cholerae* population (14, 30, 31). To gain a deeper understanding of the heterogeneity, we combine CFU counts and rates into event probabilities (Fig. 7) (26). This shows that the dynamic activity is concentrated in the last two compartments of the small intestine, the ileum sections *SI*4 and -5. In human tissue studies, *V. cholerae* was found to adhere more strongly to the epithelial cells of ileal lymphoid follicles than villi in either the jejunum or ileum, suggesting the ileum as the preferred niche in humans as well (32).

In contrast to the distal small intestine, *V. cholerae* cells are less likely to survive in the proximal small intestine. After initial colonization, bacterial numbers drop, only to increase again at later time points. Because of the low replication probabilities, this increase is likely mostly due to recolonization of this section from other sections of the small intestine, especially in *SI*1. Here, we observe that *V. cholerae* is cleared by 9 h ($N_B \leq 1$) when infected with a low inoculum, only to be recolonized by 24 h. The *SI*4 and -5 regions may serve as a source for such a recolonization process based on the high event probability for retrograde migration (Fig. 7). This observation is consistent with previous data from infant rabbit models that show that an increase in the proximal region of the small intestine is due to migration from the distal region (17). While the biological and medical

consequences of such retrograde colonization remain unclear, it is noteworthy that this effect could be shown in two independent animal models.

One advantage of our study design is that we gain information about the population structure in addition to the CFU data. We can convert this genetic information into a single measure ($N_B$) that is essential for our modeling. In addition, it also serves as an intuitive measure, the founding population size. We observe experimentally and predict with our model that $N_B$ is linearly correlated with the inoculum size and that, on average, 4.9% (95% CI, 2.4 to 6.9%) of the infectious dose initially colonize the host and are the founders of the infection. This is the hallmark of a fractional bottleneck (20), indicating that a percentage of the inoculum is able to overcome host defenses. At very high inocula, the experimental data trend toward saturation of the $N_B$, which indicates an absolute bottleneck (20) and suggests a limited availability of initial colonization niches in the host. However, our model does not predict this saturation, and the experimental data are not sufficient to distinguish between a linear or saturating fit. Unfortunately, it is technically impossible to test even higher infectious doses.

We tested our model with data sets obtained with different experimental conditions and noticed that despite a generally good fit ($R^2 = 0.84$), the model overestimates the final bacterial burden for lower inocula and under predicts for higher inocula. We therefore also investigated the effect of inoculum size on bacterial burden. Basic logistic growth models would predict that the bacterial burden depends on the inoculum size (or the founding population size) because the larger the initial populations is, the more of a head start it has to reach carrying capacity. However, the observed dependence of bacterial burden on inoculum size is much more pronounced than logistic growth models predict. In particular, we show in Fig. 6A that the experimental dose-response data can be explained by making the carrying capacity a saturating function of the infectious dose. This could be due to a cooperative effect between *V. cholerae* cells that makes the environment more permissive for the pathogen. The biological mechanisms responsible for the dose-dependent effect could, for example, be variation in the magnitude or composition of the inflammatory response or the ability of the bacterial population to overcome the innate immune response. Additionally, killing of host-associated microbiota by *V. cholerae*, via type VI secretion, is potentially contributing to competitive effects (33).

Inoculum-dependent disease progression has been reported in other diseases. For example, the bacterial disease salmonellosis and viral diseases caused by influenza A virus, severe acute respiratory syndrome coronavirus (SARS-CoV), and SARS-COV-2 cause more severe symptoms at higher exposures compared to milder and asymptomatic infections at lower doses (34–38). However, this is not universally true. In hepatitis models, a small number of viral particles leads to a chronic hepatotoxic infection, while larger inocula cause acute infections with minimal hepatotoxicity (39, 40). The correlation between disease severity and the size of the infectious dose shapes the population dynamics of pathogens and the epidemiology of infections (12).

This work suggests that it will be worthwhile to implement a prevention strategy aimed at reducing the infectious dose to mitigate the severity of disease symptoms. While this does not fully prevent disease, it might be beneficial at the population level to increase the rate of asymptomatic infections (41). Our method allows for dissecting changes in population dynamics by intervention and mutations continually over the course of disease, potentially aiding in therapeutic development. This work does not make any organism-specific assumptions and could be easily adapted to other organisms, such as viruses or eukaryotic cells, e.g., parasites and cancer cells, assuming they can be labeled with fitness-neutral markers.

## MATERIALS AND METHODS

**Modeling details.** Isogenic tagging is used to determine the rates of individual dynamic processes that govern the within-host dynamics of microbial populations (3–5, 19). Population genetic models estimate bottleneck sizes where the isogenic tag data are analyzed qualitatively (17, 20–22). In this work, we

combine stochastic population dynamic models and population genetic models to make it possible to infer the rates of individual dynamical processes from population genetic data.

The model considers a bacterial population with a total of $k$ uniquely identifiable tags in compartment $j$. In each compartment, the bacteria divide with a rate, $\beta_j$, defined as the inverse average time it takes for a bacterium to divide and die with a rate, $\delta_j$, defined as the inverse average time it takes a bacterium to die. In addition, the bacteria migrate from compartment $j$ to the nearest neighbor compartment, $j \pm 1$, with rate $\Phi_{j,j\pm1}(0)e^{-\gamma_{j,j\pm1}t}$, where the migration rate, the inverse average time it takes for a bacterium to migrate from one compartment to the next, is time dependent and exponentially declining with a relaxation constant, $\gamma_{j\pm1} > 0$. The total population size in compartment $j$ at time $t$ includes previous replication, death, and migration events, so $N_j(t)$ is a function of the replication rate, death rate, and migration rate. The initial population size at $t = 0$ is 0 for all compartments $j$ except for the stomach, where $j = 0$ (Table 1). The time evolution of the bacterial subpopulation $i$ in compartment $j$, $n_{i,j}(t)$, is assumed to be a random continuous-time Markov process where the mean subpopulation size at time $t$ is $<n_{i,j}(t)>$. Likewise, the total population size (CFU) in compartment $j$ at time $t$ is $N_j(t) = \sum_{i=1}^{k} n_{i,j}(t)$.

Hence, the proportion of cells with barcode $i$ in compartment $j$ at time $t$ is $f_{i,j}(t) = \frac{n_{i,j}(t)}{N_j(t)}$. Because of limited resources and space, a growing bacterial population reaches a carrying capacity $K_j$ in compartment $j$ in the long-time limit. Table 1 contains a summary of all the parameters in the model.

**The founder population size ($N_B$).** The founder population size, $N_B$, an approximation of the effective population size, is a quantity in population genetics and evolutionary biology that serves as a measure of the rate of change in the composition of a population due to random sampling events (17, 42). The founder population size in compartment $j$ at time $t$, $N_{Bj}(t)$, is estimated from the proportion of cells with tag $i$ in compartment $j$ at time $t$, $f_{i,j}(t)(t)$, the proportion of cells with tag $i$ in the stomach initially, $f_{i,0}(0)$, and the total number of uniquely identifiable tags, $k$,

$$NB_j^{-1}(t) = \frac{1}{k}\sum_{i=1}^{k}\frac{[f_{i,j}(t)-f_{i,0}(0)]^2}{f_{i,0}(0)[1-f_{i,0}(0)]} \tag{1}$$

The founder population size is given in terms of its inverse for notational simplicity in equation 1 (43). From the perspective of a stochastic model, the founder population size is a random variable. Therefore, the connection to stochastic population dynamics is made by taking the average on both sides of equation 1 and expanding the proportions in terms of population sizes with the aim of expressing the average founder population size in terms of the mean and the variance of the population sizes (CFU). The full derivation can be found in Text S1 in the supplemental material, where the main result is

$$\left\{ \begin{aligned} <NB_{i,j}(t)^{-1}> &\approx \frac{<n_{i,j}(t)>}{<N_j(t)>}[\frac{<n_{i,j}(t)>}{<N_j(t)>}-2f_{i,0}(0)]+\frac{\mathrm{Var}(n_{i,j}(t))}{<N_j(t)>^2}[1-4\frac{<n_{i,j}(t)>}{<N_j(t)>}+2f_{i,0}(0)] \\ &+\frac{<n_{i,j}(t)>}{<N_j(t)>^3}\mathrm{Var}[N_j(t)][3\frac{<n_{i,j}(t)>}{<N_j(t)>}-2f_{i,0}(0)]+f_{i,0}(0)^2 \\ <NB_j^{-1}(t)> &\approx \frac{1}{k}\sum_{i=1}^{k}\frac{<NB_{i,j}(t)^{-1}>}{f_{i,0}(0)[1-f_{i,0}(0)]} \\ <NB_j(t)> &\approx \frac{1}{<NB_j^{-1}(t)>} \end{aligned} \right\} \tag{2}$$

In equation 2, the auxiliary variable $<N_{Bi,j}(t)>$ is defined for notational simplicity and ease of calculations. Operationally, equation 2 states that the auxiliary variable can be calculated from knowing the initial proportions of tags and the mean and the variance of the subpopulation sizes, $n_{i,j}(t)$, and population sizes, $N_j(t)$ (equation 2, first row). Next, the inverse founder population size in compartment $j$ is determined using equation 2, second row, and the founder population size in compartment $j$ is determined using equation 2, third row.

To determine the mean and the variance of the population sizes at a particular time point and location, we use the moment-closure approach to solve a system of ordinary differential equations for the means, variances, and covariances of the population sizes given the linear compartmental topology of the model (Fig. 1B) (44). The usefulness of using the moment-closure approach to analyze isogenic tag experiments has recently been investigated from a theoretical perspective (45). One current limitation of this approach is that it becomes computationally intractable to solve for the moments for a population with $k$ on the order of $10^2$ to $10^3$ when the subpopulations are interdependent due to cooperative or competitive effects. For this reason, the founder population size is modeled by ignoring the carrying capacity, which reduces the system of dynamic processes to first order. In addition to the MC approach becoming exact in this case, an additional benefit is that only the total population size, $N_j(t)$, needs to be considered in defining and numerically solving the MC equations (44). The mean subpopulation size, $<n_{i,j}>$, and the variance of the subpopulation size, $\mathrm{Var}[n_{i,j}(t)]$, necessary to estimate the founder population size in equation 2 are obtained by multiplying the corresponding quantities for the total population size with the initial proportions, i.e., $<n_{i,j}> = <N_j(t)>f_{i,0}(0)$ and $\mathrm{Var}[n_{i,j}(t)] = \mathrm{Var}[N_j(t)(t)f_{i,0}(0)]$. The requirement for this to hold true is that the mean and the variance are directly proportional to the initial

population sizes in all compartments (Fig. S4) and, consequently, $<n_{i,j}(t)>/<N_j(t)> = f_{i,0}(0)$ and Var$[n_{i,j}(t)]/\text{Var}[N_j(t)] = f_{i,0}(0)$. Figures S5 and S6 illustrate schematics that summarize how the $N_B$s and the CFU numbers are calculated.

**The population size (CFU).** The population size is modeled using a system of ordinary differential equations for the mean population sizes, i.e., the standard deterministic approach, where the carrying capacity is taken into account using logistic growth (equation 3). Hence, the division rate and death rate are uniformly scaled with a factor that asymptotically goes to 0 as $<N_j(t)>$ approaches the carrying capacity, $K_j$, in compartment $j$ from below. The division rate and death rate are both set to 0 if the population size exceeds the carrying capacity in any of the compartments on account of using a very large infectious dose. The migration rates to nearest neighbor compartments are conjectured to be time dependent and exponentially declining functions of time, where $\Phi_{j,j\pm1}(t) = \Phi_{j,j\pm1}(0)e^{-\gamma_{j,j\pm1}t}$. Figure S1 shows a considerable improvement in the model fit and predictions for a model with time-dependent migration rates, in contrast to a less complex model with constant migration rates.

$$\frac{d<N_j(t)>}{dt} = (\beta_j - \delta_j)<N_j(t)>\left(1 - \frac{<N_j(t)>}{K_j}\right) - [\phi_{j,j+1}(t) + \phi_{j,j-1}(t)]<N_j(t)> + \phi_{j+1,j}(t)<N_{j+1}(t)>$$

$$+ \phi_{j-1,j}(t)<N_{j-1}(t)> \tag{3}$$

**Moment-closure and particle swarm optimization.** To determine the parameters in the model, we couple the moment-closure approach to particle swarm optimization (46–48). The divergence between the experimental data and the model output, $\varepsilon$, to be minimized is defined as

$$\varepsilon = \sum_{\text{Compartments}} \sum_{\text{Timepoints}} \left(\frac{<N_j(t)>_{\text{Model}} - <N_j(t)>_{\text{Experiment}}}{<N_j(t)>_{\text{Experiment}}}\right)^2 + \left(\frac{<NB_j(t)>_{\text{Model}} - <NB_j(t)>_{\text{Experiment}}}{<NB_j(t)>_{\text{Experiment}}}\right)^2 \tag{4}$$

where the sum is over the compartments corresponding to the small intestine region $SI$1 to -5. Figure S5 illustrates our heuristic PSO scheme for finding a parameter set that minimizes the error. The best-fit numerical values for the parameters are given in the supplemental material.

**Whole small intestine $N_B$ values.** To use the model to predict $N_B$ values over a larger region of the small intestine, corresponding to a subset $\Omega$ of $j = \{0,1,\ldots,6\}$, the variances and the covariances need to be appropriately summed over $\Omega$ (equation 5, second row) to get the total variance in the population size. The CFU for the coarse-grained region $\Omega$ is straight-forwardly calculated by summing the CFU number over each compartment that makes up the coarse-grained region of interest (equation 5, first row). Substituting equation 5, first and second rows, for equation 2, the founder population size ($N_B$) is calculated for the coarse-grained region $\Omega$ using $<N_\Omega(t)>$ and Var$[N_\Omega(t)]$.

$$\left\{ \begin{array}{c} <N_\Omega(t)> = \sum_{j\in\Omega} <N_j(t)> \\ \text{Var}[N_\Omega(t)] = \sum_{j\in\Omega} \text{Var}[N_j(t)] + \sum_{r\in\Omega}\sum_{l\in\Omega} \text{cov}[N_r(t), N_l(t)] \end{array} \right\} \tag{5}$$

**Strains.** A tagged library of *V. cholerae* El Tor O1 Inaba strain C6707 was constructed as previously described for this study (17). Briefly, a 1,055-bp fragment that included 93 bp of intergenic region between VC0610 and VC0611 of *V. cholerae*, and a 30-bp random sequence was inserted into the suicide plasmid pG704. The plasmid constructs were transferred to *V. cholerae* by conjugation with SM10 lambda *pir* and integrated into the genome by homologous recombination. A library of 500 individual colonies, corresponding to approximately 500 unique tags due to the possibly of sibling colonies containing the same tag or single colonies with multiple tags because they originated from more than one bacterium, were individually grown to an optical density at 600 nm ($OD_{600}$) of 0.300, concentrated to an $OD_{600}$ of 10, combined with dimethyl sulfoxide (DMSO) to 10% (vol/vol), aliquoted at 1 ml, and frozen at −80°C. Each experiment utilized a frozen aliquot produced from the same stock. The library used in this study had been previously constructed and tested to be fitness neutral (17). Growth conditions for bacteria were LB medium (number L3147 or number L3522; Miller, Sigma) supplemented with 50 $\mu$g/ml carbenicillin (number C1389; Sigma) at 37°C with broth cultures shaken at 225 rpm. Optical densities were measured at 600 nm (Genesys20; Thermo) in a 1-cm-gap cuvette (number 5510; Thermo).

**Preparation of infectious dose.** A 1-ml aliquot was thawed at room temperature and diluted 1:10 in 9 ml LB broth with 50 $\mu$g/ml carbenicillin at 37°C with shaking for 3 h ($OD_{600}$, ~0.5). Following recovery, samples were pelleted and suspended in 10 ml freshly made 300 mM sodium bicarbonate, pH 9.0, and diluted to an $OD_{600}$ of 0.143 ($2 \times 10^8$ CFU/ml). The suspension was diluted to create inoculum at the desired concentration for 50 $\mu$l per mouse dose.

**Animal experiments.** All animal protocols were approved by the Norwegian Mattilsynet (Statens Tilsyn for Planter, Fisk, Dyr, og Næringsmidler, FOTS id 8672). Six- to 7-week-old 11-day pregnant female Crl:CD1(ICR) mice were supplied by Charles River Germany. Animals were kept in specific-pathogen-free housing with one mouse per cage. Adult mice were kept until 5 days postbirth and then euthanized. Infant mice (5 days old) were separated from the dams 2 h prior to infection and randomly assigned to groups (length of infection/infectious dose). Infant mice were anesthetized by placing them in a jar atop

a platform above cloth doused in isoflurane (number FDG9623; Baxter). Gavage was performed with BD Intramedic PE tubing (0.28-mm inner diameter, 0.61-mm outer diameter; number 15250116; Fischer). Tubing was inserted 2 cm into the mouse stomach, measured from the mouth, and 50 $\mu$l was injected. McCormick green food coloring (10 $\mu$l/ml) was included in the inoculum to confirm delivery to the stomach. Mice were placed back in cages to recover.

Mice were sacrificed by decapitation and the gut excised and separated into small intestine and colon with cecum. For sectional analysis, the small intestine was evenly divided into 5 segments (~3 cm each). Samples were collected in 1.5-ml homogenizing tubes (number 525-0650; VWR) containing 1 ml phosphate-buffered saline (PBS; number P4417; Sigma) and two 2.4-mm metal beads (number 15505809; Fischer) and homogenized for 2.5 min in a Mini-Beadbeater-16 (607; Biospec Products). Samples were serially diluted in PBS and plated for counts on LB medium agar plates with 50 $\mu$g/ml carbenicillin. Counts (CFU/ml) were adjusted to tissue weight for CFU/sample; 200 $\mu$l of the homogenate was also plated on LB medium agar plates with 50 $\mu$g/ml carbenicillin and grown overnight at 37°C for STAMP analysis.

**STAMP sample processing.** Plates for STAMP analysis were harvested by placing 5 ml PBS on top of the plate and scraping. The $OD_{600}$ of a 1:10 dilution of the sample was measured, and the dilution factor was calculated for an $OD_{600}$ of 1.0. Using the calculated dilution factor, the original solution was diluted and 1 ml pelleted. Genomic DNA extraction was performed on the pellet by adding 600 $\mu$l 2% (wt/vol) sodium dodecyl sulfate, 0.5 M ethylenediaminetetraacetic acid, pH 8.0, lysis buffer for 5 min at 80°C, and then 3 $\mu$l RNase A solution (R6148; Sigma) was added for 30 min at 37°C. Cell debris was precipitated with 200 $\mu$l 7.5 M ammonium acetate and then centrifuged. DNA was precipitated from the supernatant with 800 $\mu$l of isopropanol and then washed with 70% (wt/wt) ethanol and suspended in 100 $\mu$l molecular-grade water.

Illumina MiSeq sequencing samples were generated using PCR with primers targeting the barcode flanking sequences with custom indexes and sequencing primer overhangs and manufacturer's recommended P5 to P7 regions (Table S1). PCR was performed with OneTaq 2× master mix (M0482; NEB) spiked with 1 U of Phusion high-fidelity DNA polymerase (M0530; NEB). Three 50-$\mu$l PCRs were performed per sample with 20-cycle reactions to minimize replication bias and then combined and purified using a QIAquick PCR purification kit (number 28104; Qiagen) per the manufacturer's protocol. PCR products were confirmed by gel electrophoresis and concentration determined by NanoDrop (ND-1000; ThermoFisher). Samples were combined for a concentration of 10 ng/$\mu$l each sample. The final concentration of the sample was measured by Qbit (number Q32854; ThermoFisher) and diluted to 8 nM. Sequencing was performed on an Illumina MiSeq System TruSeq HT assay per the manufacturer's protocol using MiSeq reagents kit v2, 50 cycles (number MS-102-2001; Illumina), with custom sequencing primers (Table S1).

**Code.** Codes for generating the moment-closure equations and for applying the PSO optimization scheme were implemented in MATLAB R2017b (MathWorks). All code for reproducing the data in this work is attached as supplemental material and is free to redistribute and/or modify under the terms of the GNU General Public License as published by the Free Software Foundation, either version 3 of the license, or any later version. All code is available on SourceForge (https://sourceforge.net/projects/restamp/files/WithinHostModeling/).

## SUPPLEMENTAL MATERIAL

Supplemental material is available online only.

**TEXT S1**, DOCX file, 0.03 MB.
**FIG S1**, TIF file, 0.1 MB.
**FIG S2**, TIF file, 0.1 MB.
**FIG S3**, TIF file, 0.3 MB.
**FIG S4**, TIF file, 0.1 MB.
**FIG S5**, TIF file, 0.8 MB.
**FIG S6**, TIF file, 0.9 MB.
**TABLE S1**, XLSX file, 0.01 MB.

## ACKNOWLEDGMENTS

This work was funded by Research Council of Norway (NFR) grant 262686 (to P.A.Z.W.) and 249979 (to S.A.) and Helse-Nord Grant 14796 (to S.A.).

The funders had no role in study design, data collection and interpretation, or the decision to submit the work for publication.

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
