## [Reviewer comments · mSystems]

The infectious dose shapes *Vibrio cholerae* within-host dynamics

Aaron Gillman, Anel Mahmutovic, Pia Abel zur Wiesch, and Soeren Abel

Corresponding Author(s): Soeren Abel, UiT -The Arctic University of Norway

Review Timeline:

Submission Date:	May 27, 2021
Editorial Decision:	October 6, 2021
Revision Received:	October 12, 2021
Accepted:	November 9, 2021

Editor: Vanni Bucci

Reviewer(s): Disclosure of reviewer identity is with reference to reviewer comments included in decision letter(s). The following individuals involved in review of your submission have agreed to reveal their identity: Pinaki Biswas (Reviewer #1)

Transaction Report:

DOI: <https://doi.org/10.1128/mSystems.00659-21>

October 6, 2021

Prof. Soeren Abel
The Pennsylvania State University
Department of Veterinary and Biomedical Sciences
218 Wartik
University Park, PA 16802

Re: mSystems00659-21 (The infectious dose shapes *Vibrio cholerae* within-host dynamics)

Dear Prof. Soeren Abel:

Thank you for submitting your manuscript to mSystems. We have completed our review and I am pleased to inform you that, in principle, we expect to accept it for publication in mSystems. However, acceptance will not be final until you have adequately addressed the reviewer comments.

Preparing Revision Guidelines

Sincerely,

Vanni Bucci

Editor, mSystems

Journals Department
Reviewer comments:

Reviewer #1 (Comments for the Author):

Reviewer's comments to the authors:

In the manuscript entitled "The infectious dose shapes *Vibrio cholerae* within-host dynamics" with manuscript number (mSystems00659-21), the authors studied population dynamics of *Vibrio cholerae* in mouse model using genomic tags. They also give a mathematical model that describe growth, death, forward and retrograde-migration rates during infection. After reviewing the manuscript, I have found the following concerns they need to be resolved.

Keywords

- P1L16: "Vibrio cholera". Spelling mistake.

Abstract

Introduction

- P2L46: "changing the pathogen composition". What did the authors mean by change in pathogen composition? Please clarify.
- P3L64: Correction: "have highlighted the necessity to further understand the factors".
- P4L80: "More recent developments in modeling in other pathogens have used sequence tags" grammatical error.
- P4L95: In the statement "tagged with ~500, 30 bp long, tags." is confusing. The authors should mention as "tagged with ~500 unique, 30 bp long tags." There is one more concern that why there is an approximation for 500 tags? Why there is not a fixed number of tags used in the experiment?

Results

Fitting a multi-compartmental model to experimental data

- P5L119: "inserted in a fitness neutral position in the genome." The authors should mention about the fitness neutral position in the *V. cholerae* genome where they incorporate the tag.
- P5L120: In the introduction section the authors mention "Furthermore, these data can be difficult to obtain, as their measurement often requires sacrificing the host, thereby prohibiting continuous measurements within a single animal and limiting the number of data points due to ethical considerations.". But in the result section, the authors said "sacrificed the animals at different time points (1 h, 3 h, 6 h, 9 h, and 24 h) and isolated *V. cholerae* from different sites of the gastrointestinal tract.". If the authors sacrificed different experimental animal in different time points then how did they get "continuous measurements within a single animal and limiting the number of data points" as they argued in the introduction to support their importance of study?

Figure

- Figure 1: If all the bacteria with the heritable tag of green colour is considered dead and there is no migration of any bacteria with this tag from the stomach to intestine then how come they reappear in compartments 2-5 and 6?
- Figure 3: Is there no death rate in S13 and S15?

Materials and Methods

Modeling Details

- P28L525: The authors didn't mention what ' γ ' means.
- Are the authors want to define the equation in L529 of total population size in j compartment at time t before any bacterial migration i.e., the initial population size? Or it is the total population size after migration?

Reviewer's comments to the authors:

In the manuscript entitled “The infectious dose shapes *Vibrio cholerae* within-host dynamics” with manuscript number (mSystems00659-21), the authors studied population dynamics of *Vibrio cholerae* in mouse model using genomic tags. They also give a mathematical model that describe growth, death, forward and retrograde-migration rates during infection. After reviewing the manuscript, I have found the following concerns they need to be resolved.

Keywords

- P1L16: “*Vibrio cholera*”. Spelling mistake.

Abstract

Introduction

- P2L46: “changing the pathogen composition”. What did the authors mean by change in pathogen composition? Please clarify.
- P3L64: Correction: “have highlighted the necessity to further understand the factors”.
- P4L80: “More recent developments in modeling in other pathogens have used sequence tags” grammatical error.
- P4L95: In the statement “tagged with ~500, 30 bp long, tags.” is confusing. The authors should mention as “tagged with ~500 unique, 30 bp long tags.” There is one more concern that why there is an approximation for 500 tags? Why there is not a fixed number of tags used in the experiment?

Results

Fitting a multi-compartmental model to experimental data

- P5L119: “inserted in a fitness neutral position in the genome.” The authors should mention about the fitness neutral position in the *V. cholerae* genome where they incorporate the tag.
- P5L120: In the introduction section the authors mention “Furthermore, these data can be difficult to obtain, as their measurement often requires sacrificing the host, thereby prohibiting continuous measurements within a single animal and limiting the number of data points due to ethical considerations.”. But in the result section, the authors said “sacrificed the animals at different time points (1 h, 3 h, 6 h, 9 h, and 24 h) and isolated *V. cholerae* from different sites of the gastrointestinal tract.”. If the authors sacrificed different experimental animal in different time points then how did they get “continuous measurements within a single animal and limiting the number of data points” as they argued in the introduction to support their importance of study?

Figure

- Figure 1: If all the bacteria with the heritable tag of green colour is considered dead and there is no migration of any bacteria with this tag from the stomach to intestine then how come they reappear in compartments 2-5 and 6?
- Figure 3: Is there no death rate in SI3 and SI5?

Materials and Methods

Modeling Details

- P28L525: The authors didn't mention what ' γ ' means.
- Are the authors want to define the equation in L529 of total population size in j compartment at time t before any bacterial migration i.e., the initial population size? Or it is the total population size after migration?

Reviewer's comments to the authors:

In the manuscript entitled "The infectious dose shapes *Vibrio cholerae* within-host dynamics" with manuscript number (mSystems00659-21), the authors studied population dynamics of *Vibrio cholerae* in mouse model using genomic tags. They also give a mathematical model that describe growth, death, forward and retrograde-migration rates during infection. After reviewing the manuscript, I have found the following concerns they need to be resolved.

Keywords

- P1L16: "*Vibrio cholera*". Spelling mistake.
Corrected.

Abstract

Introduction

- P2L46: "changing the pathogen composition". What did the authors mean by change in pathogen composition? Please clarify.
The "pathogen composition" was in reference to the makeup of the bacterial population at sites of infection. For clarity the wording was changed to: "Pathogens can also migrate within the host, moving to new regions of the body or changing the composition of subpopulations by migrating between colonized regions."
- P3L64: Correction: "have highlighted the necessity to further understand the factors".
Corrected. "Recent outbreaks have highlighted the need to further understand the factors critical for colonization and infection by *V. cholerae* [9,10]."
- P4L80: "More recent developments in modeling in other pathogens have used sequence tags"
grammatical error.
Corrected. "Recent developments in modeling other pathogens have used the WITS method (wild-type isogenic tag strains) to study bacterial population dynamics within the host."
- P4L95: In the statement "tagged with ~500, 30 bp long, tags." is confusing. The authors should mention as "tagged with ~500 unique, 30 bp long tags." There is one more concern that why there is an approximation for 500 tags? Why there is not a fixed number of tags used in the experiment?
Changed to "We employ a library of bacteria tagged with ~500 unique, 30 bp long tags (Material and Methods – Strains)." A reference to the materials and methods sections was added and this section was modified to clarify the approximation of tag numbers.
During creation of the barcoded bacterial population, random tags get inserted into the genome. 500 individual colonies were picked, grown up separately and the same number of bacteria from each culture was combined to create the final population. Each colony is likely, but not necessarily, formed from a single bacterium containing a single barcode; each bacterium likely (but not necessarily) contains a unique barcode. Therefore, the population contains about 500 tags. Even after sequencing the tagged population, it is difficult to accurately determine the number of unique barcodes due to sequencing noise. Fortunately, the exact number of barcodes is not important for the analysis as it is based on changes in the barcode frequencies. See also Abel et al. 2015 for more details.

Results

Fitting a multi-compartmental model to experimental data

- P5L119: "inserted in a fitness neutral position in the genome." The authors should mention about the fitness neutral position in the *V. cholerae* genome where they incorporate the tag.
We updated the material and methods section to briefly describe the library construction, the library we

used in this study was constructed in a previous study, which is cited [Abel et al, 2015]. The location in the genome is now stated with a reference for that position being fitness neutral.

- P5L120: In the introduction section the authors mention “Furthermore, these data can be difficult to obtain, as their measurement often requires sacrificing the host, thereby prohibiting continuous measurements within a single animal and limiting the number of data points due to ethical considerations.”. But in the result section, the authors said “sacrificed the animals at different time points (1 h, 3 h, 6 h, 9 h, and 24 h) and isolated *V. cholerae* from different sites of the gastrointestinal tract.”. If the authors sacrificed different experimental animal in different time points then how did they get “continuous measurements within a single animal and limiting the number of data points” as they argued in the introduction to support their importance of study?
This is a very good observation from the reviewer and we missed to opportunity to highlight that this is actually a strength of our method. While many other methods require continuous measurements within the same animal, the method this paper is based on, RESTAMP (Mahmutovic et al. 2021), is able to combine data from different animals. This enables us to take combine different time points in our model even though sampling is a terminal procedure.

Figure

- Figure 1: If all the bacteria with the heritable tag of green colour is considered dead and there is no migration of any bacteria with this tag from the stomach to intestine then how come they reappear in compartments 2-5 and 6?
That was an oversight. Good catch! We changed the colors.
- Figure 3: Is there no death rate in SI3 and SI5?
The death rates are not zero, but they are extremely low with most of the loss of CFU from those regions being a result of migrations. We added a statement clarifying this “Death rates in SI3 and SI5 were minimal, $<0.001 \text{ min}^{-1}$, with nearly all loss of bacteria in these compartments from migration.”

Materials and Methods

Modeling Details

- P28L525: The authors didn't mention what 'y' means.
We now define y directly in the main text and in Table 1.
- Are the authors want to define the equation in L529 of total population size in j compartment at time t before any bacterial migration i.e., the initial population size? Or it is the total population size after migration?
We added “The total population size in compartment j at time t includes previous replication, death and migration events, so $N_j(t)$ is a function of the replication rate, death rate and migration rate. The initial population size at $t=0$, is 0 for all compartments j except for the stomach where $j=0$ (Table 1).” to the main text for clarity.

November 9, 2021

Prof. Soeren Abel
UiT -The Arctic University of Norway
Department of Pharmacy
Universitetsvegen 57
Tromsø 9037
Norway

Re: mSystems00659-21R1 (The infectious dose shapes *Vibrio cholerae* within-host dynamics)

Dear Prof. Soeren Abel:

Your manuscript has been accepted, and I am forwarding it to the ASM Journals Department for publication. For your reference, ASM Journals' address is given below. Before it can be scheduled for publication, your manuscript will be checked by the mSystems senior production editor, Ellie Ghatineh, to make sure that all elements meet the technical requirements for publication. She will contact you if anything needs to be revised before copyediting and production can begin. Otherwise, you will be notified when your proofs are ready to be viewed.

Publication Fees:

We recognize that the video files can become quite large, and so to avoid quality loss ASM suggests sending the video file via <https://www.wetransfer.com/>. When you have a final version of the video and the still ready to share, please send it to Ellie Ghatineh at eghatineh@asmusa.org.

Sincerely,

Vanni Bucci
Editor, mSystems

Journals Department
Figure S2: Accept
Figure S6: Accept
Figure S1: Accept
Figure S4: Accept
Figure S3: Accept
Table S1: Accept
Figure S5: Accept
Text S1: Accept